# IPAQ-L and CPET Usefulness in a North-Eastern Romanian Population Undergoing Cardiac Rehabilitation

**Andrei Manta [1], Elena Cojocaru [2,3] , Maria Magdalena Leon-Constantin [1,4,\*], Alexandra Maștaleru [1,4,\*], Mihai Roca [1,4], Cristina Rusu [3,5], Sabina Alexandra Cojocariu [1,4] and Florin Mitu [1,4]**

[1] Department of Medical Specialties (I), Faculty of Medicine, "Grigore T Popa" University of Medicine and Pharmacy, University Street nr 16, 700115 Iasi, Romania; andrei.manta@d.umfiasi.ro (A.M.); mihai.c.roca@umfiasi.ro (M.R.); alexandra-sabina_o_cojocariu@d.umfiasi.ro (S.A.C.); florin.mitu@umfiasi.ro (F.M.)

[2] Department of Morphofunctional Sciences I, "Grigore T. Popa" University of Medicine and Pharmacy, 700115 Iaşi, Romania; elena2.cojocaru@umfiasi.ro

[3] Saint Mary Emergency Medicine Children Hospital, Vasile Lupu Street nr 62, 700309 Iasi, Romania; cristina.rusu@umfiasi.ro

[4] Clinical Rehabilitation Hospital–Cardiovascular Rehabilitation Clinic, Pantelimon Halipa Street nr 14, 700661 Iasi, Romania

[5] Department of Mother and Child Medicine, "Grigore T. Popa" University of Medicine and Pharmacy, University Street nr 16, 700115 Iaşi, Romania

\* Correspondence: maria.leon@umfiasi.ro (M.M.L.-C.); alexandra.mastaleru@umfiasi.ro (A.M.)

**Abstract:** (1) Background: Current guidelines emphasize the importance of regular moderate and/or high intensity aerobic exercises in cardiovascular disease prevention. Our study aimed to evaluate the utility of the International Physical Activity Questionnaire Long Form (IPAQ-L) for its physical activity (PA) quantification in patients with heart failure with reduced ejection fraction. (2) Methods: We conducted a cross-sectional study of 110 patients aged between 34 and 69 years admitted to the Cardiovascular Rehabilitation Clinic. All patients underwent a clinical examination, blood tests, a cycle ergometer exercise stress test and individual assessment of their weekly PA level using the IPAQ-L. (3) Results: Obesity, hypertension and type 2 diabetes were highly prevalent in our study group but did not influence the IPAQ-L results. In terms of physical performance, moderate intensity was the most common level of intensity found in our study group. Regarding the data on the relationship between the IPAQ-L questionnaire and cardiopulmonary exercise testing (CPET) parameters, vigorous PA was correlated with predicted maximal oxygen uptake ($p = 0.025$) and moderate PA, in addition to walking, were correlated with heart rate reserve ($p = 0.005$ and $p = 0.009$, respectively). (4) Conclusions: IPAQ-L can be used for the evaluation of individual PA levels within a cardiovascular rehabilitation program, but cannot substitute for the importance and utility of CPET.

**Keywords:** cardiovascular rehabilitation; heart failure; hypertension; obesity; cardiopulmonary exercise testing; physical activity

## 1. Introduction

### 1.1. Background and Rationale

Sedentarism, defined as deficient physical activity for more than three months, is one of the most important modifiable cardiovascular risk factors [1] and a key component of every cardiovascular rehabilitation (CR) program. The current European Guidelines [2] recommend at least 150 min of moderate intensity aerobic physical activity per week, 75 min of vigorous intensity aerobic physical activity per week or an equal combination of moderate and vigorous intensity activities. The guidelines suggest that effort should be distributed across multiple sessions of at least 10 min each. Heart failure with reduced ejection fraction (HFrEF) is characterized by significant exercise intolerance caused primarily by skeletal muscle atrophy and dysfunction [3]. The efficacy and safety of exercise-based

cardiac rehabilitation were documented for the first time by Coats et al. [4]. Recently, a meta-analysis that evaluated the usefulness of the program in patients with heart failure showed an increase in quality of life and exercise capacity, but did not find a significant advantage in terms of mortality and hospitalization [5].

In the absence of physical activity, as in the current COVID-19 pandemic especially, obesity increases significantly and a multidisciplinary approach is essential. Numerous actions must be applied to increase physical activity and decrease obesogenic lifestyles, thus determining a long-term decrease in morbidity and mortality [6]. A study published in 2020 suggested that the variations in moderate to vigorous physical activity and sedentary activity evaluated through the International Physical Activity Questionnaire (IPAQ) or through accelerometry can help define a population's health profile. Furthermore, in overweight/obese adults with hypertension, a 16 week supervised aerobic exercise program was successful in increasing self-reported physical activity, lowering sedentarism and enhancing sleep quality [7].

The development of techniques to accurately assess physical activity (PA) has essential importance for both research and follow-up of CR patients, who frequently struggle with poor adherence to lifestyle changes. PA duration, frequency and intensity should be taken into account in a comprehensive assessment, making such a measurement complex and challenging [1]. Several methods that measure PA have been proposed: direct observation, patient diaries, PA questionnaires and direct measurement [8]. Self-report questionnaires and physical activity monitors are the most common techniques used in the CR program [1].

Despite recent significant advancements in portable and direct physical activity monitors, clinical evidence regarding their reliability in cardiovascular disease patients is still scarce [9]. The monitor must be worn 12 h per day for at least 4 days to produce an acceptable PA assessment. Although it has the benefit of quantifying inactivity periods, such a system is still unavailable for the majority of Romanian middle-aged and elderly adults, who constitute the majority of patients admitted to Romanian Cardiac Rehabilitation Clinics, making self-report questionnaires a better method for such facilities [1,9]. Furthermore, even if these devices are available on the Romanian market—at inaccessible prices for patients—the Clinical Rehabilitation Hospitals in Romania do not have the economic potential to offer these devices to patients. In Romania, CR programs are not funded separately and there are also no policies through which cardiovascular patients are sent to this service, so there are minimal opportunities to address the issue.

The major limitation of self-report questionnaires is the patient's ability to remember and categorize their recent PA as easy, moderate and vigorous. The IPAQ questionnaire provides comparable estimations of PA and it has been validated in 14 centers across 12 countries. IPAQ comes in two versions. While the Short Form is recommended for large prevalence studies, the IPAQ Long Form (IPAQ-L) provides the advantage of a more detailed analysis of PA across four different domains (occupation, transportation, home and leisure time) [10]. The IPAQ-L addresses PA performed over the previous seven days, assessing, for each of the four domains listed above, the frequency and average time of walking, moderate activity and vigorous activity.

### 1.2. Study Objectives

Our hypothesis was that the IPAQ-L questionnaire would be useful for patients from Romania, given its accessibility and low cost. The primary objective of the study was to evaluate the usefulness of the IPAQ-L questionnaire in patients with HFrEF included in a CR program. The secondary objectives of the study were: (i) the evaluation of the relationship between the items of the IPAQ-L questionnaire and the clinical, biological and paraclinical characteristics of the patients; (ii) the influence of cardiovascular comorbidities, including, obesity, hypertension and type 2 diabetes, on IPAQ-L results; and (iii) the relationship between the PA parameters evaluated by the IPAQ-L questionnaire and the effort capacity determined by CPET for a suitable subgroup. The usefulness of the IPAQ-L evaluation compared to CPET lies in the capacity to assess the criterion validity for a

PA questionnaire using a direct method. Validation using a direct method is needed to estimate the absolute amount of PA and is most relevant when monitoring adherence to health-enhancing PA recommendations.

## 2. Materials and Methods

### 2.1. Study Design and Setting

We performed a single-center cross-sectional study among patients aged 18–69 years admitted between 1 January 2017 and 31 December 2017 in the Cardiovascular Clinic of the Rehabilitation Clinic Hospital in Iasi, Romania, with the aim of assessing PA patterns in patients with HFrEF.

### 2.2. Ethics Approval

The clinical study obtained approval from the local Ethics Committee for Scientific Research of the University of Medicine and Pharmacy of Iasi (certificate of approval: 5483/8 March 2017) and the Ethics Committee for Scientific Research of the Rehabilitation Clinic Hospital (certificate of approval from 27 January 2017). All patients signed an informed consent form and then underwent a clinical examination, blood tests and a cycle ergometer stress test and provided self-evaluation of their PA level using the IPAQ-L questionnaire.

### 2.3. Study Population

The study was a single-center, cross-sectional study, including 110 adult patients with HFrEF who adhered to a CR program and signed the written informed consent form. The diagnostic criterion of HFrEF during the admission was a left ventricular ejection fraction less than or equal to 40%, a hemodynamic parameter measured with transthoracic echocardiography. A total of 298 hospitalized patients with the HFrEF diagnosis were screened for their eligibility to participate in the study. Of these, 141 of them were diagnosed with NYHA IV heart failure, 14 were diagnosed with psychological or cognitive impairment that limited the CR, 12 patients had locomotive disorders that excluded them from participation in an exercise training program, 4 individuals undertook PA for more than 7 h per day and/or 28 h per week and 17 were not interested in participating in the study. Finally, 110 patients with HFrEF were eligible to be included the study. All of the above can be seen in Figure 1.

### 2.4. Study Procedures and Outcome Assessment

Patient assessment included medical history (comorbidities, clinical information, anthropometrics); blood tests, such as lipid profile; and cardiovascular and pulmonary evaluation. PA assessment was conducted using the IPAQ-L, which was presented within the first 24 h of admission. The English version of the questionnaire was translated into Romanian, following current guidelines and recommendations [11]. The interview was conducted by a single physician trained to deal with patient interrogation. The total time spent performing different types of PA was converted into minutes. As short training sessions are known to have insignificant metabolic effects [12], the final analysis of PA level included only activities with a minimum duration of 10 min. Physical activity levels were expressed as the total metabolic equivalent of task (MET)-minutes/week, obtained by multiplying predefined MET scores by the duration of a specific PA (in minutes) [13]. We calculated the total MET-minutes/week, as well as the value for each activity (walking, moderate effort and vigorous effort) and for each domain (work, transportation, leisure and domestic and garden activities), respectively. Furthermore, following IPAQ scoring guidelines [14], we divided our study group in three categories based on patients' PA levels: low, moderate and high. Sitting time/week was reported as the total time in minutes. Sedentary time was quantified as minutes/week.

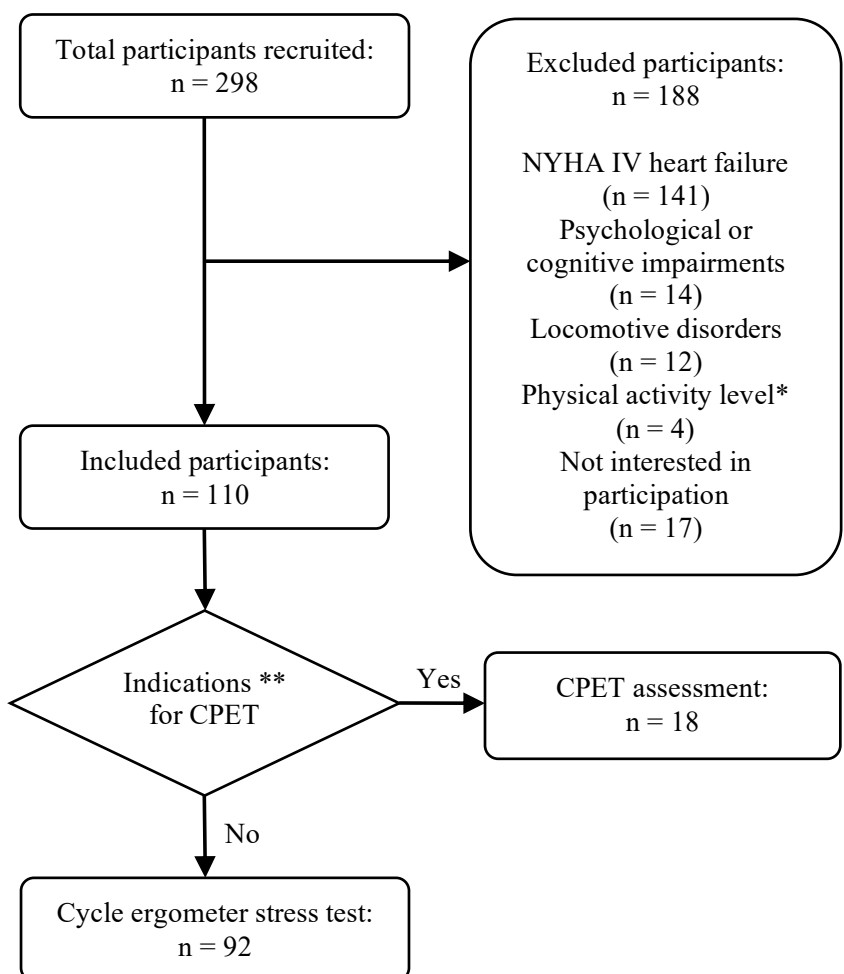

* Physical activity level more than 16 hours / day and / or 28 hours / week (IPAQ-L questionnaire)
** Confirmation of the cardiovascular pathology predominant contribution in exercise capacity decreasing, for cases associating non-cardiovascular comorbidities, which also result in functional capacity alteration

**Figure 1.** Study flow chart.

Utilizing body mass index (BMI), our study group was categorized as normal weight (18.5–24.9 kg/m$^2$), overweight (25–29.9 kg/m$^2$), first-degree obesity (30–34.9 kg/m$^2$), second-degree obesity (35–39.9 kg/m$^2$) and third-degree obesity (>40 kg/m$^2$).

The abdominal circumference (AC) was measured at the midpoint of the line between the rib or costal margin and the iliac crest in the midaxillary line. Abdominal obesity was defined as a waist circumference >88 cm for females and >102 cm for males.

Fitness was quantified by the percentage of age-predicted maximal heart rate (%HR), exercise resistance (W) and workload (METS) through a symptom-limited cycle ergometer stress test using standard protocol [15].

Patients in whom the etiology of exercise capacity limitation could not be clearly established were judged suitable for cardiopulmonary exercise testing (CPET). A subgroup of 18 patients was thus formed and this investigation was carried out on the same day as the PA assessment with IPAQ-L, a few hours after the patients completed it. The most important CPET parameters were: the absolute value of maximal oxygen uptake (VO$_2$ max) and the percentage of this predicted value (VO$_2$ max%), the absolute value of the maximal work rate (WR) and the percentage of this predicted value (WR%), the oxygen uptake at the anaerobic threshold (AT), the maximal value of the respiratory exchange ratio (RER), the maximal heart rate (HR) and the heart rate reserve (HRR). The HRR is determined by the difference between the maximal HR and the resting HR.

*2.5. Statistical Analysis*

All statistical analyses were carried out with SPSS v 20.0 (SPSS Inc., Chicago, IL, USA), using the chi-square test and Student's t-test for comparisons between groups. The Mann–Whitney U test was used as an alternative to Student's t-test when the data were not normally distributed. Descriptive data were displayed as means $\pm$ the standard deviation (SD), medians with interquartile range or percentages, as appropriate. A *p*-value < 0.05 was considered statistically significant. Correlations between variables were assessed by calculation of Spearman's correlation coefficients.

## 3. Results

*3.1. Patient Characteristics*

Our study population included 110 patients with an average age of 57.2 years. The descriptive statistics of our study population are illustrated in Table 1. The group exhibited a balanced gender ratio (47.27% males and 52.72% females), with similar BMI and blood pressure values upon admission. Although the male subgroup presented a higher average glycemia than the female population, females had a lower lipid profile than males. Regarding the cycle ergometer stress test results, females achieved a slightly higher %HR, but male subjects presented both better exercise resistance (105.84 W versus 75.84 W) and a higher workload (5.21 METS versus 4.54 METS), as described in Table 1.

**Table 1.** Descriptive statistics of the study population.

| Variables | Total | Males | Females | *p*-Value * |
|---|---|---|---|---|
| Number, *n* (%) | 110 (100) | 52 (47.27) | 58 (52.72) | |
| Age (years), mean (SD) | 57.20 (6.45) | 57.11 (7.05) | 57.27 (5.92) | 0.897 |
| Weight (kg), mean (SD) | 83.48 (12.40) | 88.15 (12.41) | 79.29 (10.89) | <0.001 |
| BMI (kg/m$^2$), mean (SD) | 30.28 (4.80) | 29.66 (5.22) | 30.84 (4.36) | 0.200 |
| AC (cm), mean (SD) | 97.81 (12.41) | 101.09 (13.18) | 94.87 (10.96) | 0.008 |
| SBP (mmHg), mean (SD) | 134.44 (15.31) | 132.96 (17.57) | 135.77 (12.97) | 0.346 |
| DBP (mmHg), mean (SD) | 83.65 (10.52) | 83.88 (10.20) | 83.44 (10.89) | 0.829 |
| Glycemia (mg/dL), mean (SD) | 114.73 (38.90) | 121.61 (35.79) | 108.45 (32.49) | 0.048 |
| TC (mg/dL), mean (SD) | 200.00 (42.97) | 190.15 (45.68) | 208.82 (38.68) | 0.022 |
| HDL-C (mg/dL), mean (SD) | 48.22 (13.15) | 45.17 (11.87) | 51.05 (13.75) | 0.019 |
| LDL-C (mg/dL), mean (SD) | 121.88 (37.99) | 115.98 (39.76) | 127.45 (35.73) | 0.127 |
| Non-HDL cholesterol (mg/dL), mean (SD) | 151.39 (41.00) | 144.98 (44.14) | 157.35 (37.26) | 0.118 |
| TG (mg/dL), mean (SD) | 158.72 (99.80) | 161.09 (111.53) | 156.60 (88.91) | 0.815 |
| EGFR (mL/min/1.73 m$^2$), mean (SD) | 84.02 (16.58) | 84.02 (17.28) | 81.34 (15.60) | 0.075 |
| LVEDD (mm), mean (SD) | 48.28 (6.29) | 51.63 (5.14) | 45.08 (5.61) | <0.001 |
| LVMI (g/m$^2$), mean (SD) | 120.96 (35.58) | 134.11 (33.44) | 107.19 (32.74) | <0.001 |
| %HR (%), mean (SD) | 75.34 (12.54) | 72.82 (11.89) | 77.60 (12.77) | 0.045 |
| Exercise resistance (W), mean (SD) | 90.02 (30.17) | 105.84 (30.01) | 75.84 (22.38) | <0.001 |
| Workload (METS), mean (SD) | 4.85 (1.33) | 5.21 (1.39) | 4.54 (1.20) | 0.008 |
| Weight status, *n* (%): Normal weight | 10 (9.09) | 6 (11.54) | 4 (6.9) | 0.512 |
| Overweight | 52 (47.27) | 25 (48.08) | 27 (46.50) | 1.000 |
| First-degree obesity | 30 (27.27) | 15 (28.85) | 15 (25.86) | 0.831 |
| Second-degree obesity | 15 (13.63) | 5 (9.62) | 10 (17.24) | 0.278 |
| Third-degree obesity | 3 (2.72) | 1 (1.92) | 2 (3.45) | 1.000 |
| Hypertension, *n* (%): Normotensive | 22 (20) | 15 (28.85) | 7 (12.07) | 0.033 |
| First-degree HTN | 10 (9.09) | 2 (3.85) | 8 (13.79) | 0.099 |
| Second-degree HTN | 20 (18.18) | 5 (9.62) | 15 (25.86) | 0.046 |
| Third-degree HTN | 58 (52.72) | 30 (57.69) | 28 (48.28) | 0.345 |
| Type 2 diabetes, *n* (%) | 31 (28.18) | 21 (40.38) | 10 (17.24) | 0.010 |

*: Mann–Whitney U test; BMI: body mass index; AC: abdominal circumference; SBP: systolic blood pressure; DBP: diastolic blood pressure; TC: total cholesterol; HDL-C: high density lipoprotein cholesterol; LDL: low density lipoprotein cholesterol; non-HDL: non-high-density lipoprotein cholesterol; TG: triglycerides; EGFR: estimated glomerular filtration rate; LVEDD: left ventricular end-diastolic diameter, LVMI: left ventricular mass index; %HR: percentage of age-predicted maximal heart rate; HTN: hypertension.

Overweight, first-degree, second-degree and morbidly obese patients accounted for 47.27%, 27.27%, 13.63%, and 2.72% of our study group, respectively. Among the study participants, 80% were hypertensive and more than 50% were diagnosed with third-degree hypertension (Table 1). The presence of hypertension and/or diabetes did not influence the IPAQ-L results. Obese and overweight patients required a longer time to accomplish the walking parameter compared to the subjects with a body mass index less than 25 kg/m$^2$ ($\Delta$ = 2306.56 MET-minutes/week, $p$ < 0.001 and $\Delta$ = 1765.81, $p$ < 0.001, respectively), but no significant difference was found regarding total weekly PA among these subgroups.

Moderate PA was the most common activity level for both males and females, accounting for over 60% of total PA. Females were more likely to be involved in physical activities, and 31.79% of males had a low PA rate (Table 2).

**Table 2.** Distribution of total physical activity among subgroups.

|  | Total | Males | Females |
|---|---|---|---|
| Low level of PA (%) | 27.07 | 31.79 | 22.40 |
| Moderate level of PA (%) | 65.35 | 62.62 | 68.04 |
| High level of PA (%) | 7.57 | 5.58 | 9.54 |

PA: physical activity.

There were no statistically significant differences between genders in terms of weekly PA, nor for any of the four analyzed domains or the difficulty levels (walking, moderate exercise and vigorous exercise), as shown in Table 3. The domestic and garden domain was the preferred type of PA for both males and females, accounting for approximately 60% of total MET-minutes per week.

**Table 3.** IPAQ questionnaire results.

|  | Age | | MTCF | |
|---|---|---|---|---|
|  | *R*-Value | *p*-Value * | *R*-Value | *p*-Value * |
| Total MET-minutes/week at work | −0.28 | 0.002 | 0.28 | 0.002 |
| Total MET-minutes/week for transportation | −0.01 | 0.850 | 0.01 | 0.850 |
| Total MET-minutes/week in domestic and garden activities | −0.01 | 0.852 | 0.01 | 0.852 |
| Total MET-minutes/week in leisure time | 0.04 | 0.631 | −0.04 | 0.631 |
| Vigorous PA (leisure time) | −0.12 | 0.183 | 0.12 | 0.183 |
| Total physical activity MET-minutes/week | −0.18 | 0.053 | 0.18 | 0.053 |
| BMI (kg/m$^2$) | 0.15 | 0.107 | −0.15 | 0.107 |
| AC (cm) | 0.10 | 0.295 | −0.10 | 0.295 |
| LVEF (%) | 0.10 | 0.338 | −0.10 | 0.338 |

*: Spearman's rho. MET: total metabolic equivalent of task; MTCF: maximum tolerated cardiac frequency; PA: physical activity; BMI: body mass index; AC: abdominal circumference; LVEF: left ventricular ejection fraction.

### 3.2. Outcomes

Age was significantly correlated with total MET-minutes/week at work (r = −0.28, $p$ = 0.002), but not with the total activity in the other three analyzed domains or with sitting time ($p$ > 0.05). Furthermore, age presented a weak borderline correlation with total physical activity MET-minutes/week (r = −0.18, $p$ = 0.053), but not with BMI, AC or EF. Maximum tolerated cardiac frequency (MTCF) fitness exhibited a significant positive correlation with total MET-minutes/week at work (r = 0.28, $p$ = 0.002), and a borderline correlation with total physical activity MET-minutes/week (r = 0.18, $p$ = 0.053), but not with BMI or AC (Table 4).

**Table 4.** Correlations between age and MTCF and physical activity, BMI, AC and LVEF.

| | Total Mdn (IQR) | (%) | Males Mdn (IQR) | (%) | Females Mdn (IQR) | (%) | *p* Value * |
|---|---|---|---|---|---|---|---|
| Total MET-minutes/week at work | 6132 (1936–11,370) | 21.1 | 15,210 (6194–25,524) | 22.6 | 2866 (2087–6936) | 19.7 | 0.384 |
| Total MET-minutes/week for transportation | 594 (198–1386) | 18.1 | 988 (181–8752) | 20.3 | 877 (330–4788) | 15.9 | 0.517 |
| Total MET-minutes/week in domestic and garden activities | 2520 (240–5790) | 54.5 | 12,675 (4147–17,085) | 50.8 | 1350 (180–6720) | 58.1 | 0.019 |
| Total MET-minutes/week in leisure time | 198 (33–594) | 6.1 | 123 (49–198) | 6.1 | 579 (99–1482) | 6.1 | 0.983 |
| Total sitting (minutes) | 540 (420–660) | | 545 (300–692) | | 540 (360–600) | | 0.899 |
| Total walking (MET-minutes/week) | 1072 (346–2227) | 27.1 | 5131 (1249–10,065) | 31.7 | 2095 (280–5841) | 22.4 | 0.799 |
| Total moderate activity (MET-minutes/week) | 2840 (415–8115) | 65.3 | 13,575 (11,797–17,085) | 62.6 | 5340 (1620–7920) | 68.0 | 0.210 |
| Total vigorous activity (MET-minutes/week) | 2880 (780–7440) | 7.5 | 7440 (4800–19,800) | 5.5 | 2400 (600–4320) | 9.5 | 0.057 |
| Total physical activity (MET-minutes/week) | 4735 (1614–12,515) | | 29,382 (23,833–37,727) | | 14,061 (3080–20,091) | | 0.964 |

*: Mann–Whitney U test; MET: total metabolic equivalent of task; Mdn: median; IQR: interquartile range.

Vigorous activity was statistically significant correlated with $VO_2$ max% (r = 0.52, *p* = 0.025) and with AT (r = 0.53, *p* = 0.026). However, the correlation between $VO_2$ max% and total physical activity did not reach statistical significance (r = 0.34, *p* = 0.168). Total physical activity presented significant correlations with WR% (r = 0.48, *p* = 0.04) and HRR (r = 0.65, *p* = 0.003), as determined by CPET. HRR also correlated with vigorous activity (r = 0.61, *p* = 0.007), moderate activity (r = 0.63, *p* = 0.005), and walking (r = 0.59, *p* = 0.009) (Table 5).

**Table 5.** Correlations between CPET parameters and IPAQ-L questionnaire.

| CPET Parameters | IPAQ-L and Physical Activity (METS-Minutes/Week) | | | | | | | |
|---|---|---|---|---|---|---|---|---|
| | Vigorous Activity | | Moderate Activity | | Walking | | Total Physical Activity | |
| | R-Value | *p*-Value | R-Value | *p*-Value | R-Value | *p*-Value | R-Value | *p*-Value |
| $VO_2$ max% | 0.52 | 0.025 | 0.19 | 0.447 | 0.32 | 0.195 | 0.34 | 0.168 |
| AT | 0.53 | 0.026 | 0.12 | 0.632 | 0.16 | 0.522 | 0.30 | 0.228 |
| RER | 0.02 | 0.923 | 0.18 | 0.468 | 0.16 | 0.502 | 0.07 | 0.781 |
| WR% | 0.52 | 0.027 | 0.45 | 0.061 | 0.48 | 0.040 | 0.48 | 0.040 |
| HRR | 0.61 | 0.007 | 0.63 | 0.005 | 0.59 | 0.009 | 0.65 | 0.003 |

$VO_2$ max%: percentage of the predicted maximal oxygen uptake; AT: oxygen uptake at the anaerobic threshold; RER: maximal value of the respiratory exchange ratio; HRR: heart rate reserve.

## 4. Discussion

The IPAQ questionnaires have been studied in industrialized countries and in the urban population of developing countries. However, when interpreting IPAQ findings from rural or low-literacy communities in developing countries, caution has been advised [8]. Although the IPAQ-L was validated for PA monitoring among adults [8], it lacks accuracy, like most self-reported methods of evaluating PA levels, as patients frequently overestimate their level of PA [16]. The IPAQ-L in particular yields an overestimation of moderate to vigorous PA and an underestimation of sitting time [17,18]. However, since a recent study demonstrated that patients interviewed by a trained professional reported PA levels much

closer to the real ones than those who filled in the IPAQ questionnaire by themselves [17], we decided to use this approach in our evaluation.

Although it has been reported that the IPAQ-L is associated with greater overestimation of total PA compared to the IPAQ Short Form [10], its advantage consists in its capacity to differentiate domain-related PA, offering a detailed picture of the study group's activity patterns. Such data may be useful in the creation of effective intervention programs to combat sedentary behavior [19]. Despite the fact that other authors have reported a negative correlation between physical activity levels and BMI [20], we did not find any significant correlation between BMI or AC and weekly PA in the analyzed domains.

In 2019, Lee et al. showed that males exert a greater amount of moderate to vigorous daily effort (mostly occupational and leisure-related) while females spend less time performing PA, and it is generally in household-related areas [20]. Although it has been reported that gender substantially influences the amount and type of PA, both males and females from our study group exerted similar physical effort, as assessed via the IPAQ questionnaire. In our study, moderate PA was the most common activity form for both genders and, although males reported a higher number of weekly METS than females, the difference did not achieve statistical significance. Females performed more moderate activities inside their homes and the total number of METS reported in the domestic and garden domain was not statistically significant. The difference between the amount of vigorous PA performed by males versus females in the domestic and garden domain reached borderline correlation.

The time spent on leisure activities as evaluated by the IPAQ is considered to be the most consistent compared to the other domains [21]. Leisure time physical activity exhibits a north-to-south decline in the European Union, ranging from 24 MET-hours/week in Sweden to less than 10 MET-hours/week in southern countries such as Portugal, Spain, Italy and Greece. The average physical leisure activity time in our study group was 512.14 ($\pm$1049.75) MET-minutes/week, similar to other southern European countries [21,22].

The utility of IPAQ-L in adults with type 2 diabetes, with and without peripheral neuropathy, has been studied by Nolan et al. [23]. The authors demonstrated that patients with type 2 diabetes and peripheral neuropathy were significantly less active than people with type 2 diabetes alone ($p = 0.04$). A recent study comparing direct assessment of PA with IPAQ and indirect assessment of PA with accelerometry in patients with type 2 diabetes showed that there was no significant difference between the two methods ($p < 0.05$). Therefore, their findings indicate that the IPAQ may serve as a potential tool for PA assessment in patients with type 2 diabetes [24].

In patients with hypertension, Riegel et al. [25] showed a low agreement between self-reporting of adherence to PA in the clinical setting and in IPAQ interviews. Moreover, the authors noted that the PA recommendation has a low association with BP control in the clinical setting, suggesting that medical advice alone is not able to translate the effectiveness of supervised PA demonstrated in clinical trials to clinical practice. For sedentary hypertensive females, Bravo et al. [26] showed that self-reported PA using IPAQ was predominantly related to domestic ($p = 0.018$) and work activities ($p = 0.001$) at moderate intensity. Therefore, IPAQ appears to be an adequate instrument to assess the energy expenditure of hypertensive patients and its impact on their aerobic capacity.

Although previous studies have found an age-related decrease in PA [27,28], the correlation only reached a borderline value in our analysis. While several authors have previously reported a natural inverse relation between physical activity and excess weight [22,29], other studies have found an uncommon positive association between total PA and BMI (especially among females), which could be explained by health consciousness or apparent motivation [30–33]. Fan et al. [33] reported a strong negative association between middle age (40–49 years) and total PA in their study group, but also failed to show a substantial association between BMI and total PA. Although a recent study [33] showed that male gender and the 30–39-year-old age group are associated with a higher total sitting time,

we found no significant differences regarding sitting time between males and females or within the three analyzed age groups.

CPET is a clinical method that allows for a global assessment of cardiorespiratory function in order to determine exercise capacity. It allows objective measurement of both submaximal and peak exercise responses using measures of respiratory oxygen uptake, carbon dioxide output and ventilatory measures. Previous reports highlight that the CPET is often used prior to major surgery to assess functional potential in patients and aid clinical decision-making and risk assessment [34]. Moreover, a recent study assessing the utility of CPET in patients recovering from COVID-19 suggested it as a potentially useful method for detecting ventilatory and cardiovascular changes in COVID-19 [35]. Therefore, for subjects admitted to a cardiovascular and pulmonary rehabilitation program, CPET can be useful as a screening device for exercise capacity and cardio-ventilatory limitations. According to our results, the association of IPAQ-L with CPET could offer valuable information in specific populational groups.

A previous study reported a moderate correlation between IPAQ Short Form results and treadmill stress test performance [36]. Our analysis showed that MTCF as a measure of fitness level exhibited a significant positive correlation with total MET-minutes/week at work and a borderline correlation with total physical activity MET-minutes/week. However, a study of a larger population is required to confirm the latter correlation.

According to a recent study, IPAQ was not appropriate for assessing PA in cardiovascular disease patients because the subjects often recorded severe PA values, resulting in non-homogeneous outcomes [37]. Although the IPAQ questionnaire can provide a comprehensive image of a patient's PA pattern on an individual basis, we cannot suggest using IPAQ-L in the study of a wider community of subjects with various cardiovascular disease due to the broad variability of the responses reported by our patients (similar to those of Fournier et al. [37]). Although other authors have used the IPAQ questionnaire in the analysis of PA among Romanian students [38–40] and in the geriatric population [41], to our knowledge, this is the first analysis of PA levels using the IPAQ-L and CPET in Romanian patients with cardiovascular comorbidities.

## 5. Limitations of the Study

It is important to highlight the limitations of the study. First of all, we can note the small number of patients that underwent the CPET (18 participants); however, despite this limited subgroup, we found some statistically significant correlations between the parameters evaluating PA and those evaluating functional capacity through the CPET. Another drawback was the lack of published studies involving patients who have undergone CPET and have also been evaluated using IPAQ-L; we were unable to find any data focused on patients who have been admitted to a CR program. Another limitation was the fact that the patients were evaluated without an evaluation of the effect of PA over time. Moreover, this was a cross-sectional study that did not allow us to explain causal relationships, our results being descriptive ones that cannot explain biological links.

## 6. Conclusions

IPAQ-L is useful for the evaluation of individual PA levels within a CR program. IPAQ-L is a suitable tool for measuring PA in order to develop public health policy recommendations or optimize public health interventions at a very low cost. Nowadays, in the current pandemic situation, public health services are under great pressure. It would thus be helpful to monitor the PA of patients that need CR in the outpatient department through the IPAQ-L questionnaire.

Regarding the secondary outcomes, the IPAQ-L results in the patients with HFrEF were characterized by opposite values and high variability. Obesity, hypertension and type 2 diabetes were highly prevalent in our study group but did not influence the IPAQ-L results. The data from this study regarding the relationship between the IPAQ-L questionnaire and the CPET parameters were encouraging. Thus, while vigorous physical activity

was correlated with VO$_2$ max%, moderate physical activity and walking were correlated with HRR. However, the questionnaire cannot substitute for the importance of CPET in the assessment of effort performance.

With regard to future perspectives, this study opens up new horizons for further research, for which larger groups of patients would be required to certify the applicability of this tool in assessing patients with HFrEF. Moreover, the IPAQ-L questionnaire can be considered a possible instrument for use in patients with HFrEF who cannot perform cycle ergometer stress tests but are candidates for CR.

**Author Contributions:** Conceptualization, A.M. (Andrei Manta) and F.M.; methodology, E.C.; software, M.R.; validation, A.M. (Andrei Manta); investigation, S.A.C.; resources, A.M. (Alexandra Maștaleru); writing—original draft preparation, A.M. (Alexandra Maștaleru); writing—review and editing, C.R.; visualization, E.C.; supervision, M.M.L.-C. All authors contributed equally. All authors have read and agreed to the published version of the manuscript.

**Funding:** This research received no external funding.

**Institutional Review Board Statement:** The study was conducted according to the guidelines of the Declaration of Helsinki and was approved by the Institutional Review Board (or Ethics Committee) of the University of Medicine and Pharmacy "Grigore T. Popa" Iasi (protocol code 5483 from 08 March 2017).

**Informed Consent Statement:** Informed consent was obtained from all subjects involved in the study. Written informed consent was obtained from the patients to publish this paper.

**Data Availability Statement:** The data presented in this study are available on request from the corresponding author.

**Conflicts of Interest:** The authors declare no conflict of interest.

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
