# Peer review of "IPAQ-L and CPET Usefulness in a North-Eastern Romanian Population Undergoing Cardiac Rehabilitation"

_applsci, doi:10.3390/app11125483_

Round 1

Reviewer 1 Report

The authors have performed a IPAQ-L questionnaire on a HFref patient population to evaluate physical activity. Overall, the author document considerable variation of IPAQ-L outcomes within the study population, which was not directly related to co-morbidities such as hypertension, obesity and diabetes. Probably this observation reflects the overall heterogeneity of the IPAQ-L results in the study population, making it very hard to draw any solid conclusion. Finally, for a limited number of patients (n=18) the authors have performed correlation of IPAQ-L and CPET outcomes, showing some degree of correlation of vigorous activity with VO2max% and overall with heart rate reserve.

Overall, the manuscript is well written and structured, although mainly descriptive. However, I am missing a final conclusion and critical appraisal of the relevance of this work for clinical practice. Moreover, did the authors define upfront criteria to objectively define "the usefulness" or "equivalency"of IPAQ-L evaluation compared to CPET. 

While the authors have a solid dataset of IPAQ-L in HFref patients, they fail to convey a convincing and clear-cut message with relevance for clinical practice. In part, this might be since the outcome (/correlation) is rather disappointing and maybe not reflecting the initial hypothesis that IPAQ-L would be a helpful tool for stratification of HFref patients (maybe to the very high heterogeneity of the population). Therefore, I would recommend the authors further reflect on the objectives and key conclusions of their study. 

Author Response

Dear reviewer,

Thank you for all your comments.

We have tried to explain better in the objectives section the usefulness of IPAQ-L evaluation compared to CPET. We have also added in the conclusions section a critical appraisal of the relevance of this work for clinical practice.

Hope we have touched all the points you asked us to change.

If there are any other changes you consider we should make, please let us know.

Yours sincerely,

All the authors

Reviewer 2 Report

Thank you for opportunity to review the paper titled IPAQ - L and CPET usefulness in a North-Eastern Romanian 2 population undergoing cardiac rehabilitation . The work is well written and clarifies the results clearly and consciously. In addition it is a relevant subject for clinicians and researchers. The methods are apropriate. Flow is very good. Only one point- 65- 68 Are the portable direct physical activity monitors are available in other countries? The data would be usuful.

Author Response

Dear reviewer,

Thank you for all your comments.

We have described in the background section the Romanian situation and policies regarding these portable devices and the possibility of offering them to the patients.

Hope we have touched all the points you asked us to change.

If there are any other changes you consider we should make, please let us know.

Yours sincerely,

All the authors

Reviewer 3 Report

This manuscript describes a cross-sectional study in which International Physical Activity Questionnaire Long Form (IPAQ - L) and cardiopulmonary exercise testing (CPET) were used to evaluate the physical activity of the heart failure patients from one center in northeastern Romania. This study has some novelty for the use of IPAQ - L in the physical activity assessment in the heart failure patients and the newly-found relationship between IPAQ - L results and CPET parameters. In the following, I listed several major concerns need to be addressed.

  1. This paper talks about IPAQ - L and CPET evaluation in patients with heart failure with reduced ejection fraction. Because heart failure is basically a progressive process, how to well explain the statement “cardiac rehabilitation” in title and conclusion?
  2. Is the sample size of patients with CPET large enough to detect a clinically significant result? Please add a paragraph about sample size and power estimation.
  3. In the part of study design, the authors need to clearly define the timing of the CPET the patients accepted. Because the cross-sectional study can only show the prevalence of a condition on one specific time point, the CPET should be carried out soon after the IPAQ – L test in order to build a convincing relationship between the results from the two tests.
  4. The authors enrolled patients with HFrEF into their study. However, the authors did not clearly define the diagnostic criteria of HFrEF during the admission. A recruitment standard needs to be added into the method part.
  5. In Table 1, a p value is needed between males and females for each variables.
  6. It seems that the authors thought short training had insignificant metabolic effects, however, it needs citations to support.
  7. There is a typo in 3.3 study population part. It cannot be 27 hours /day.
  8. In statistical analysis part, “All statistical analyses were carried out using SPSS v 20.0” should be “All statistical analyses were carried out with SPSS v 20.0”.
  9. In statistical analysis part, “Descriptive data were expressed as” should be “Descriptive data were displayed as”.
  10. All the p in “p value” of the literature part should be italic.
  11. In 3.1 patient characteristics part, “obese patients represented 47.27 %, 27.27 %, 13.63 %, and respectively 2.72 % of our study group” should be “obese patients accounted for respectively 47.27 %, 27.27 %, 13.63 %, and 2.72 % of our study group”.
  12. In 3.1 patient characteristics part, “Between study participants, 80 % of them were hypertensive” should be “Among the study participants, 80 % of them were hypertensive”.
  13. In 3.1 patient characteristics part “p= 0.00 and Δ= 1765.81, p= 0.00, respectively”, p value cannot equal to 0.
  14. It’s improper to use “borderline” negative correlation or “borderline” significant correlation.
  15. Unify the gender expression to male/female. There should not be both men/women and male/female in one paper.

Author Response

Dear reviewer,

Thank you for all your comments.

  1. We have tried to explain better the relationship between heart failure with reduced ejection fraction, physical activity, and cardiac rehabilitation in the background section. Thus, we have tried to motivate the usage of this term in the title and conclusions, given the fact that the patients included in this study adhered to a cardiac rehabilitation program in the hospital setting.
  2. We applied CPET only in cases free of contraindication. The sample size of the subgroup with CPET is definitely too small to result in an elevated level of statistical power. We presented this issue in the “Limitation of the study”. However, despite the limited CPET subgroup, we found some statistically significant correlations, between CPET parameters defining functional capacity and physical activity assessed by IPAQ-L. We consider these correlations should be further confirmed in statistically more powerful clinical studies, including larger groups of participants.
  3. We have added in the Study procedures and outcome assessment subsection the moment when the CPET was performed compared to the IPAQ-L. Thank you for the remark.
  4. We have added in the Study population subsection the diagnostic criteria for HFrEF. We hope we have improved the recruitment standard.
  5. Thank you very much for your comment. We have added a new column in Table 1 and reported the p-value between males and females.
  6. We have added a citation from the latest guideline of Secondary prevention through comprehensive cardiovascular rehabilitation in which the authors recommend that trainings should start from at least 15 minutes and increase gradually to 30, 45 and 60 minutes for significant metabolic effects.
  7. In the study population subsection, we had typo and it is not 27 h/day, it is 7 hours/day. Thank you very much for your observation.
  8. We have changed in the statistical analysis subsection “All statistical analyses were carried out with SPSS v 20.0” as you recommended.
  9. We have changed in the statistical analysis subsection “Descriptive data were displayed as” as you recommended.
  10. We have made all the p from the p-value with italic, as you recommended. Thank you!
  11. In the patient characteristics subsection, we have changed the sentence with “obese patients accounted for respectively 27 %, 27.27 %, 13.63 %, and 2.72 % of our study group” as you recommended.
  12. In the patient characteristics subsection, we have changed the sentence with “Among the study participants, 80 % of them were hypertensive” as you recommended. Thank you!
  13. It was a typo when transferring the data from SPSS to word. Thank you very much for your observation. We have changed this in the text with p<0.001.
  14. We have made the changes you recommended.
  15. We have unified the gender expression and everywhere in the text, we have only male/female.

Thank you again for all your advice!

Hope we have touched all the points you asked us to change.

If there are any other changes you consider we should make, please let us know.

Yours sincerely,

All the authors

Round 2

Reviewer 1 Report

In my opinion, the manuscript has not significantly increased. Moreover, the revised sections contain a number of language errors.

The overall novelty and added value is rather limited and the key message/narrative remain not well developed. In fact, it remains unclear to me what the initial hypothesis was and whether a sample size of n=18 would be sufficient to draw any meaningful conclusion on the usefulness of IPAQ-L. It is imperative that the authors clearly define the study objective and success criteria.   

Author Response

Dear reviewer,

Thank you for all your comments.

We have observed and changed the language errors from the revised sections. Thank you for your remark.

We have restructured the study objectives into initial hypothesis, primary and secondary outcome. Hopefully, this will increase the clarity of our study. Also, we have reorganized the paper conclusions so they maintain the same order with the objectives and respond to them. By this, we did our best to increase the manuscript quality and improve with a convincing and clear-cut message with relevance for clinical practice

Regarding the sample size of n=18, patients in whom the etiology of exercise capacity limitation could not be clearly established were suitable for cardiopulmonary exercise testing. This fact was added in the paper for a better understanding. The sample size of the subgroup with CPET is definitely too small to result in an elevated level of statistical power. We presented this issue in the “Limitation of the study”. However, despite the limited CPET subgroup, we found some promising relations, between CPET parameters defining functional capacity and physical activity assessed by IPAQ-L. We consider these correlations should be further confirmed in statistically more powerful clinical studies, including larger groups of participants.

Hope we have touched all the points you asked us to change.

If there are any other changes you consider we should make, please let us know.

Yours sincerely,

All the authors

Reviewer 3 Report

This manuscript describes a cross-sectional study in which International Physical Activity Questionnaire Long Form (IPAQ - L) and cardiopulmonary exercise testing (CPET) were used to evaluate the physical activity of the heart failure patients from one center in northeastern Romania. This study puts up forward a new idea for local practice. I appreciate the authors highlight the realistic condition in local, which contributes to the necessity of this study. Overall I admire their efforts to offer extra low-cost options for local healthcare.

Author Response

Dear reviewer,

Thank you for all your helpful comments and for your appreciation.

Yours sincerely,

All the authors